# Repeated noninvasive stimulation of the ventromedial prefrontal cortex reveals cumulative amplification of pleasant compared to unpleasant scene processing: A single subject pilot study

**Constantin Winker[1,2‡], Maimu A. Rehbein[1,2‡], Dean Sabatinelli[3], Markus Junghofer[1,2]***

**1** Institute for Biomagnetism and Biosignalanalysis, University of Muenster, Muenster, Germany, **2** Otto Creutzfeldt Center for Cognitive and Behavioral Neuroscience, University of Muenster, Muenster, Germany, **3** Department of Psychology and BioImaging Research Center, University of Georgia, Athens, Georgia, United States of America

‡ These authors are joint first authors on this work.
* markus.junghoefer@uni-muenster.de

## Abstract

The ventromedial prefrontal cortex (vmPFC) is a major hub of the reward system and has been shown to activate specifically in response to pleasant / rewarding stimuli. Previous studies demonstrate enhanced pleasant cue reactivity after single applications of transcranial direct current stimulation (tDCS) to the vmPFC. Here we present a pilot case study in which we assess the cumulative impact of multiple consecutive vmPFC-tDCS sessions on the processing of visual emotional stimuli in an event-related MEG recording design. The results point to stable modulation of increased positivity biases (pleasant > unpleasant stimulus signal strength) after excitatory vmPFC stimulation and a reversed pattern (pleasant < unpleasant) after inhibitory stimulation across five consecutive tDCS sessions. Moreover, cumulative effects of these emotional bias modulations were observable for several source-localized spatio-temporal clusters, suggesting an increase in modulatory efficiency by repeated tDCS sessions. This pilot study provides evidence for improvements in the effectiveness and utility of a novel tDCS paradigm in the context of emotional processing.

## Introduction

The prioritized processing of emotional information within early (<100 ms) [1] or mid-latency (<300 ms) [2] time intervals strongly supports the idea that environmental emotional stimuli are of high relevance to humans. One important aspect of emotion processing is the differentiation between aversive / unpleasant and rewarding / pleasant information. In this context, the ventromedial prefrontal cortex (vmPFC) is known to play an important role. More precisely, the vmPFC shows increased activation patterns specifically for stimuli with pleasant content in comparison to unpleasant or neutral stimuli [3–8]. Furthermore, several

**Data Availability Statement:** All MEG data and results files are available from the OSF database (URL: osf.io/8e6rf).

**Funding:** This work was funded by the Interdisciplinary Center for Clinical Research of the University of Muenster (Ju2/024/15) and the Collaborative Research Center (SFB-TRR58/C08). The funders had no role in study design, data collection and analysis, decision to publish, or preparation of the manuscript.

**Competing interests:** The authors have declared that no competing interests exist.

psychiatric disorders were shown to be linked—amongst others—to dysfunctional processing in vmPFC regions [9–13].

To investigate the valence specificity of vmPFC and its potential to be modulated noninvasively, several studies within our lab were conducted that assessed neuronal effects of excitatory and inhibitory transcranial direct current stimulation (tDCS). In an fMRI and a separate MEG sample, participants showed valence modulation effects for passively viewed emotional scenes with increased neuronal activation for pleasant compared to unpleasant scenes after excitatory vmPFC-tDCS and vice versa after inhibitory vmPFC-tDCS [14]. Comparable effects were found during the processing of happy compared to fearful faces [15]. In addition, several studies from other groups point to a more general modulation capability of vmPFC by tDCS that is not limited to the mere viewing of emotional stimuli [16–20]. All these studies investigated effects of a single tDCS session. However, there is evidence that multiple sessions of tDCS might induce cumulating effects and increase efficacy [21], although others even reported cancelation of effects with a second stimulation after 24 h [22,23]. In terms of clinical application, brain stimulation techniques like tDCS or transcranial magnetic stimulation (TMS) are typically conducted across repeated sessions. For example, TMS treatment of major depressive disorder is recommended by Perera and colleagues [24] over a period of 4–6 weeks with five daily sessions per week. In the case of tDCS, reviews about safety aspects [25] as well as therapeutic use [26] propose repeated sessions, too, to increase efficacy. However, due to additional variance introduced by repeated sessions it might be recommendable to test new brain stimulation protocols to ascertain the absence of unwanted side effects or null effects (e.g. due to cancelation).

To probe the limits and expandability of our newly developed tDCS paradigm, in this pilot study we investigated the possible cumulative effects of tDCS across 5 days of consecutive sessions with author MJ as single participant. Therefore, the event-related paradigm was similar to the MEG assessment described by Junghofer and colleagues [14], featuring a passive viewing task with pleasant and unpleasant emotional scenes during MEG measurement of neuronal activation. With respect to a planned extension of this study to healthy controls, a further goal was to identify possible negative physiological or psychological side effects of repeated stimulation such as headaches, dysphoric- or depression-like symptoms during or after repeated inhibitory, or euphoria or mania-like symptoms during or after repeated excitatory stimulation. Weighing false positive less critical than false negative findings of detrimental side effects, we decided against a blinding of tDCS conditions, i.e. with distinct knowledge of potential side effects we expected a lower detection threshold. Since negative side effects of repeated inhibitory stimulation could be masked by potentially long-lasting positive side effects in the aftermath of repeated excitatory stimulation, we decided to start with repeated inhibitory stimulation.

Consistent with our prior research, we hypothesized an induction of a positivity bias after excitatory vmPFC-tDCS with increased brain activation for pleasant compared to unpleasant stimuli and vice versa after inhibitory vmPFC-tDCS by comparing neural activation after vmPFC-tDCS (Post-tDCS) with baseline activity (Pre-tDCS). Furthermore, in the case of accumulating tDCS effects, both effect patterns (positivity bias due to excitatory, negativity bias due to inhibitory tDCS) were predicted to intensify across sessions. Finally, the course of Pre-tDCS data was separately investigated to test for any cumulative effects reflecting changes of vmPFC baseline activation (i.e. increasing positivity or negativity biases of baseline activity across days). Based on our previous findings on emotional scenes [14], we expected multiple effects to occur in spatio-temporal clusters distributed across the entire interval of analysis (0–600 ms after stimulus onset). However, due to the exploratory nature of this study, analyses reported here were conducted without predefined regions of interest. Only temporal

limitations by fixed intervals that divided the whole time window into subgroups (for a detailed description see [14,15]) were applied.

## Materials and methods

### Participant

MJ (male; age: 48 years) has given written informed consent (as outlined in PLOS consent form) to publish these case details.

### tDCS

tDCS was applied by means of a DC Stimulator Plus (NeuroConn). The procedure was identical to previous studies [14,15,27]: in a two-electrode design with a 3 x 3 cm square electrode placed at 10–20 electrode position Fp and a 5 x 5 cm square electrode placed below the chin as extracephalic reference, a current strength of 1.5 mA was applied over 10 min. Both electrodes were covered in saline soaked sponges to enable sufficient conductibility.

### Stimuli

Stimuli employed in the MEG measurement have been reported previously [14]. They consisted of pleasant (erotica, romantic couples, happy children and families) and unpleasant scenes (graphic bodily injuries, threatening animals, threatening humans) with 32 different stimuli per category. Each stimulus contained a grayscale picture with 1024 x 786 resolution (12.3˚ horizontal field of view). Stimuli were controlled for luminance, contrast, and complexity (90% JPEG file size) across categories.

### Procedure

The study consisted of 10 daily sessions of Pre-tDCS and Post-tDCS MEG measures summing up to 20 MEG data sets. Participant MJ first passed five sessions of inhibitory stimulation and a month later, five sessions of excitatory stimulation on five consecutive weekdays each. The Post-tDCS MEG measurement always followed the tDCS in direct succession (<5 min). During MEG assessment, a passive viewing task was conducted. MJ relaxed and kept his gaze fixated on a central red dot to reduce artifacts from eye-movement. Stimuli were presented in a pseudo-random order with controlled transitions between scenes from Pleasant and Unpleasant categories which did not repeat until the complete set of 64 individual scenes had been presented. Overall, 192 stimuli (3 x 64 scenes) were presented per MEG run. Each of the 20 MEG measurements featured the same stimuli in newly pseudo-randomized orders. A single stimulus was presented for 600 ms followed by a jittered inter-trial-interval of 1,000–2,000 ms showing a gray background.

### MEG measurement and analysis

MEG measurements were conducted with a 275 whole-head sensor system (CTF Systems) with first-order axial gradiometers. Head movement and position were controlled via landmark coils positioned on the nasion and in both earlobes. MEG data were assessed with a sampling rate of 600 Hz. Afterwards, data were down-sampled (300 Hz) and filtered with a 0.1 Hz high-pass filter (zero-phase second-order Butterworth) and a 48 Hz low-pass filter (zero-phase fourth-order Butterworth). Trials were split into 800 ms epochs ranging from -200 to 600 ms relative to stimulus onset. Additionally, trials were baseline-adjusted by subtracting the mean of a -150 to 0 ms interval. In succession, source activation was calculated by Minimum-Norm estimates (L2-MNE) [28]. As a source model, a spherical shell with evenly distributed 2 x 350

dipoles in azimuthal and polar directions was applied. L2-MNE topographies were established with a Tikhonov regularization parameter of $k = 0.1$.

Statistical analysis comprised correlational testing of the L2-MNE data. We analyzed data with regard to three hypothesized effect patterns. Hypothesis 1: we tested if previous findings of an induced positivity bias after excitatory stimulation and negativity bias after inhibitory stimulation could be replicated and remained constant across sessions, i.e. without any cumulative effect. Therefore, the pattern of an identical activation increase per day from Pre-tDCS to Post-tDCS for difference [Pleasant minus Unpleasant] within the excitatory stimulation condition and activation decrease per day within the inhibitory stimulation condition was analyzed. We thus applied differential weights to calculated differences [Pleasant minus Unpleasant] to test for this effect (Pleasant-Minus-Unpleasant-Excitatory-Pre1-5: 2, 2, 2, 2, 2; Pleasant-Minus-Unpleasant-Excitatory-Post1-5: 3, 3, 3, 3, 3; Pleasant-Minus-Unpleasant-Inhibitory-Pre1-5: 2, 2, 2, 2, 2; Pleasant-Minus-Unpleasant-Inhibitory-Post1-5: 1, 1, 1, 1, 1). Hypothesis 2: to investigate a cumulative stimulation effect across sessions, double differences {[Pleasant minus Unpleasant] minus [Post-tDCS minus Pre-tDCS]} were assumed to increase linearly across all five sessions within the excitatory tDCS condition, whereas inhibitory stimulation was assumed to show a linear decrease across sessions. As we were interested in cumulative Post-tDCS effects, these data were to be baseline-adjusted first for each day to control for varying Pre-tDCS activations, e.g. by additional habituation effects across days (Pleasant-Minus-Unpleasant-Post-Minus-Pre-Excitatory1-5: 1, 2, 3, 4, 5; Pleasant-Minus-Unpleasant-Post-Minus-Pre-Inhibitory1-5: -1, -2, -3, -4, -5). To control for this variance, we accepted the trade-off of losing statistical power by calculating the double difference. Hypothesis 3: we finally investigated if this cumulative effect was already visible at Pre-tDCS measurements prior to daily tDCS sessions. Therefore, a correlational analysis of differences [Pleasant minus Unpleasant] at baseline (Pre-tDCS) tested for an increase of the Pleasant > Unpleasant effect across all five sessions of excitatory tDCS and vice versa (i.e. Pleasant < Unpleasant effect) across all five sessions of inhibitory tDCS (Pleasant-Minus-Unpleasant-Excitatory-Pre1-5: 5, 6, 7, 8, 9; Pleasant-Minus-Unpleasant-Inhibitory-Pre1-5: 5, 4, 3, 2, 1). Prevention of $\alpha$-error inflation was established by the use of a non-parametric cluster permutation [29]. Thereby, clusters containing $r$-values cumulated across dipole-positions and time-points (spatio-temporal cluster) were tested against a distribution of 1,000 random permutations of the assessed data. In order to add an effect to a cluster, the respective spatio-temporal point had to surpass a $p$-value of .05 (sensor-level criterion). Subsequently, all effects passing this threshold were added up to a so-called cluster mass (in this case summation of all $r$-values) and tested against the randomly permuted data. If the cluster mass reached a $p$-value < .05 in comparison to the biggest cluster of each of the 1,000 permutations (cluster-level criterion), it surpassed the critical cluster mass and thus was considered significant. Cluster mass-analyses were conducted for previously determined intervals (see [14,15,27]) of 0–100 ms (early), 100–200 ms (early mid-latency), 200–300 ms (late mid-latency) and 300–600 ms (late). First- and second-level significant clusters that were cut off by the beginning or end of a pre-defined interval were reanalyzed in an interval extended for 50 ms in the respective direction. By this means, it was possible to assess if the found cluster actually began or ended with the borders of the pre-defined interval or if it was temporally extended. Preprocessing and analysis of the MEG data was conducted using EMEGS software (emegs.org) [30].

## Results

All in the following reported significant results reflect positive $r$-scores, thus confirming the respective hypotheses described above. Reported $p$-values refer to cluster-level criterion.

Importantly and further replicating findings of our previous research [14, 15], there were no spatio-temporal clusters with inverted effect patterns with *p*-values < .2.

### Constant effects of stimulation across sessions

With regard to a constant (non-cumulative) stimulation effect across sessions (Hypothesis 1), three clusters were observed at mid-latency time intervals: in right occipitoparietal cortex (153–257 ms; *p* = .01), medial prefrontal cortex (mPFC; 167–280 ms; *p* = .002), and in right anterior temporal cortex (227–307 ms; *p* = .007). Additionally, the analysis yielded a fourth cluster during late latency in vmPFC (350–470 ms; *p* = .005) (see Fig 1).

### Cumulative effects of stimulation: Post-pre differences

Analysis of the cumulative effects of stimulation across sessions (Hypothesis 2) revealed a later mid-latency cluster at mPFC regions (217–277 ms; *p* = .001) (see Fig 2).

### Cumulative effects of stimulation: Pre differences

The investigation of a cumulative effect across Pre-tDCS measurements (Hypothesis 3) yielded three significant clusters. A first cluster was observed during early mid-latency in primary regions of occipital cortex (127–217 ms; *p* = .022). A second cluster appeared during late mid-latency in secondary occipitotemporal and occipitoparietal regions (230–293 ms; *p* = .001), while a third cluster occurred in a late interval in central parietal cortex (323–383 ms; *p* = .037) (see Fig 3).

## Discussion

In this pilot study, we investigated how repeated vmPFC-tDCS across 5 days affects the previously reported modulation of valence specificity [14,15,27].

An analysis testing for consistent effects across sessions with increased activation for pleasant stimuli compared to unpleasant stimuli after excitatory (i.e. positivity bias) and vice versa after inhibitory stimulation (i.e. negativity bias) yielded three mid-latency (right occipitoparietal, mPFC, and right anterior temporal) and one late spatio-temporal cluster (right vmPFC). These relatively stable effects of directed modulation across five daily sessions do not support strong habituation or even cancelation effects of day-to-day carryover as for instance reported by Monte-Silva and coworkers [22,23].

In contrast, we identified cumulative effect patterns due to tDCS repetitions. A significant cluster appeared at mPFC, which spatially and temporally overlapped with the cluster in Fig 1B. Thus, while the earlier activation pattern (~170–220 ms) revealed a main effect of stimulation only, the later part (~220–280 ms) suggested increasing positivity and negativity biases across days in this area.

In the same vein, another series of cumulative effect analyses investigated changes of baseline activation across days. Here, another three clusters (mid-latency: occipital cortex; occipitotemporal and occipitoparietal cortex; late latency: central parietal cortex) also supported cumulative effects consistent with our hypotheses, as pleasant stimulus reactivity showed increasing activation in comparison to unpleasant stimulus reactivity after excitatory stimulation and vice versa after inhibitory stimulation across days. An explanation for these cumulative effects could be a shift in the baseline activation level of vmPFC or the network that is affected by it. These effects support the often proposed activation of neuroplastic changes (see [26]) and by that point to a possibility to increase efficacy of this paradigm of repeated stimulations.

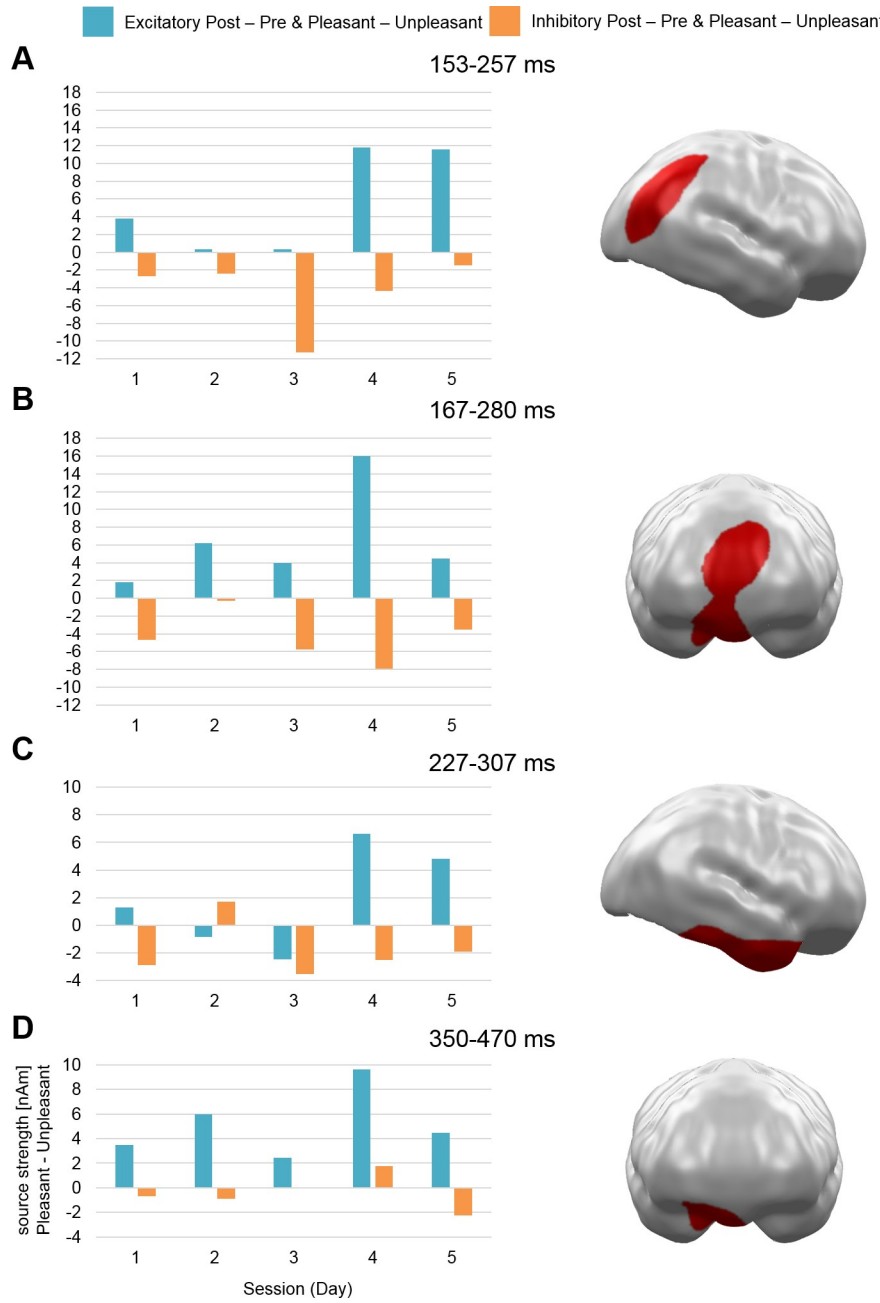

**Fig 1. Spatio-temporal clusters with stable stimulation by Valence effects across sessions (Hypothesis 1).** All clusters reflect a pattern of stronger activation for pleasant compared to unpleasant stimuli after excitatory and vice versa after inhibitory vmPFC-tDCS across all five sessions. Effects were found in (A) right occipitoparietal cortex, (B) mPFC, (C) right anterior temporal cortex, and (D) vmPFC. mPFC = medial prefrontal cortex, vmPFC = ventromedial prefrontal cortex.

Through the course of both weeks of tDCS, MJ reported mild signs of disturbed sleep with more than usual nocturnal wake-ups that both times dissolved with the end of the tDCS. No other signs of physiological or psychological side effects could be reported neither during the active stimulation nor in the weeks of stimulation or in the weeks after stimulation.

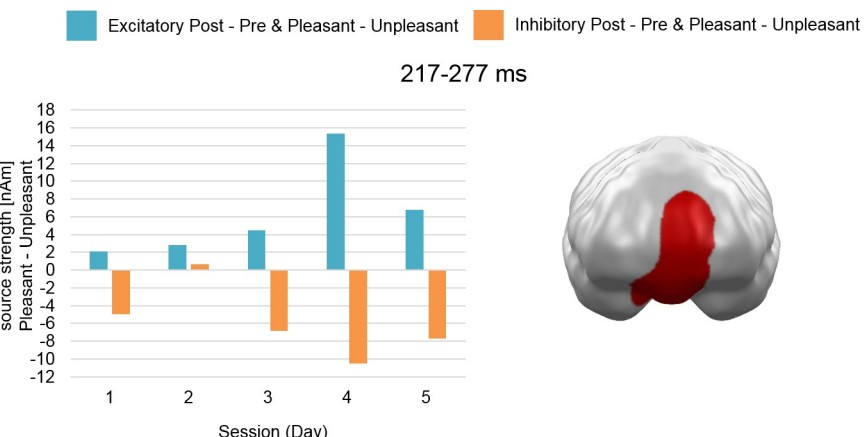

**Fig 2. Spatio-temporal cluster with cumulative stimulation by Valence effect across sessions (Hypothesis 2).** The cluster reflects an effect pattern of increasingly stronger activation for pleasant compared to unpleasant stimuli after excitatory and vice versa after inhibitory vmPFC-tDCS across all five sessions at the mPFC. mPFC = medial prefrontal cortex.

## Limitations

It is noteworthy that all spatio-temporal clusters revealed the proposed direction of effects but differed with regard to temporal and spatial characteristics from previous findings [14]. These differences might reflect inter-individual differences in brain activation patterns and brain anatomy. However, effects can be interpreted as typical emotion related activation patterns like the early posterior negativity (EPN) and the late positive potential (LPP) that have been localized in occipitotemporal as well as parietal regions [31,32]. Nevertheless, these pilot findings need to be tested in a representative sample to see if previously presented effects are actually cumulative or if the here presented effects are at least partly distinct from those.

Due to the intentional non-blinding of MJ to the tDCS conditions, it is possible that focused top-down attention—although unintentional—might have increased valence specific processing as for instance shown for explicit directed attention by Schupp and coworkers for emotional scenes [33] or Schindler and Kissler [34] for emotional words. It is also possible that implicit, unintentional reappraisal processes have influenced scene valence perception. For instance, Li and colleagues [35] presented results of successful upregulation of pleasant stimulus material. With regard to the cumulative effects, a study of Denny and Ochsner [36] pointed to possible additive effects of downregulation of negative emotions over the course of four sessions. However, in that study, two reappraisal methods were compared and only one showed cumulative effects that were not attributable solely to habituation. It therefore might require a clear regulation strategy in order to cumulatively downregulate emotions over sessions. MJ, in contrast, passively viewed all presented images without explicitly directing attention to one specific category and without applying any specific strategy. Nevertheless, as in our previous studies, follow-up studies should keep the participants blind with regard to underlying hypotheses and stimulation conditions. Furthermore, a comparison of conditions excitatory, inhibitory, and sham stimulation should be considered. On the one hand, the comparison between both active stimulation conditions helps further understanding the underlying mechanisms of tDCS and on the other hand the application of a sham group enables to differentiate placebo effects from stimulation effects.

In this pilot study, we also checked for potential physiological or psychological negative side effects of the repeated stimulation, but side effects—especially on cognitive processes—

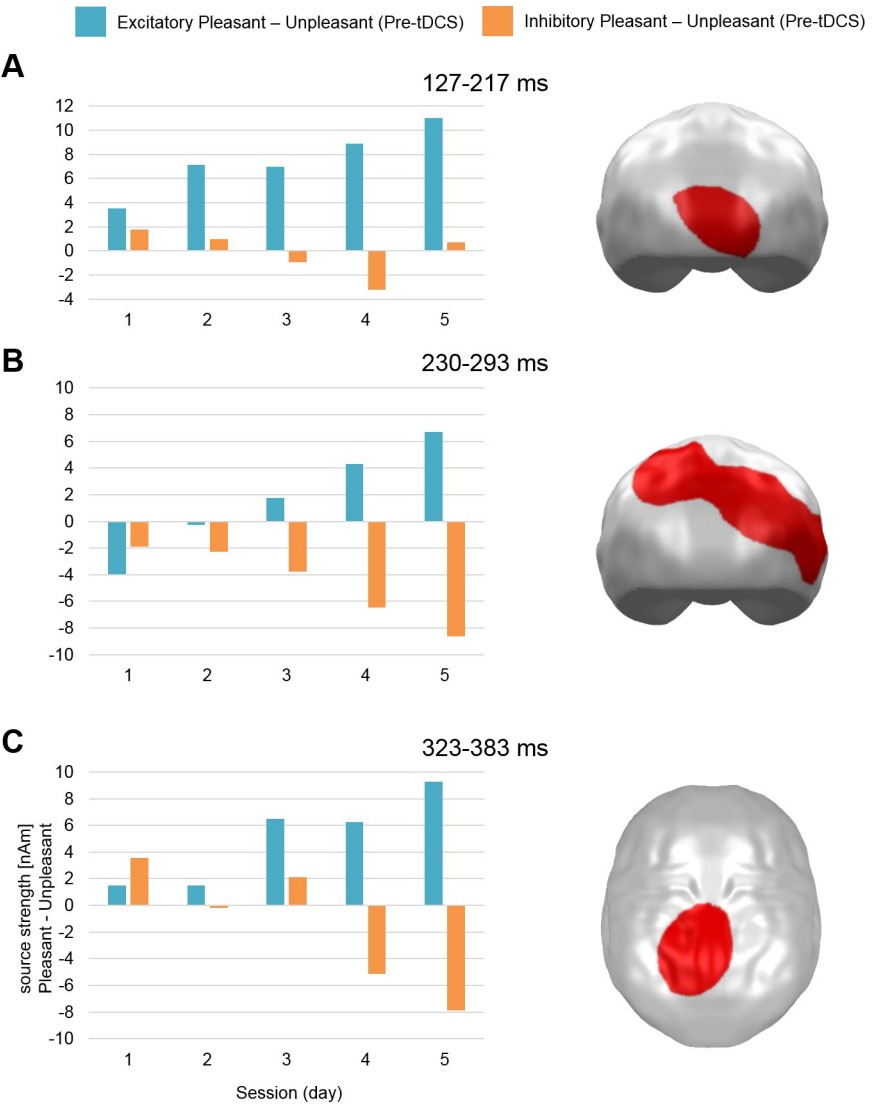

**Fig 3. Spatio-temporal clusters with cumulative stimulation by Valence effects during MEG baseline measurement before stimulation (Pre-tDCS) across sessions (Hypothesis 3).** Clusters reflect an effect pattern of increasingly stronger Pre-tDCS activation for pleasant compared to unpleasant stimuli after excitatory and vice versa after inhibitory vmPFC-tDCS across all five sessions in (A) occipital cortex, (B) occipitotemporal and occipitoparietal cortex, and (C) central parietal cortex.

have not systematically been assessed. Therefore, while this study does not speak against repeated tDCS in healthy controls, a first test of a single tDCS repetition on two consecutive days and a systematic assessment of side effects and symptoms of stimulations appears recommendable.

## Conclusion

This pilot study provides a first indication of cumulative effects of repeated tDCS on the valence processing of emotional stimuli. These cumulative effects were also visible in the MEG baseline measurement around 24 h after the preceding stimulation and thus in the absence of acute aftereffects. Repeated stimulations did not lead to more than mild negative side effects.

Of course, these results should be viewed only as preliminary as only one person participated in this study. Nevertheless, due to the absence of more systematic investigations of tDCS paradigms in the context of emotion processing, this pilot data might help to stimulate further research in this field.

## Ethics Statement

This study was approved by the Ethics Committee of the Department of Psychology and Sports Science at the University of Muenster, Germany.

## Acknowledgments

We thank Karin Wilken, Hildegard Deitermann, and Ute Trompeter for their help with the MEG data collection, and Andreas Wollbrink for technical assistance.

## Author Contributions

**Conceptualization:** Markus Junghofer.

**Formal analysis:** Markus Junghofer.

**Investigation:** Constantin Winker.

**Methodology:** Markus Junghofer.

**Software:** Constantin Winker, Markus Junghofer.

**Supervision:** Dean Sabatinelli, Markus Junghofer.

**Validation:** Constantin Winker, Maimu A. Rehbein.

**Visualization:** Constantin Winker.

**Writing – original draft:** Constantin Winker, Maimu A. Rehbein, Dean Sabatinelli, Markus Junghofer.

**Writing – review & editing:** Constantin Winker, Maimu A. Rehbein, Dean Sabatinelli, Markus Junghofer.

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
