## [Decision Letter · Decision Letter 0]

14 Oct 2019

PONE-D-19-23160

Repeated noninvasive stimulation of the ventromedial prefrontal cortex reveals cumulative amplification of pleasant compared to unpleasant scene processing: a single subject pilot study

PLOS ONE

Dear PhD Junghoefer,

Thank you for submitting your manuscript to PLOS ONE. After careful consideration, we feel that it has merit but does not fully meet PLOS ONE’s publication criteria as it currently stands. Therefore, we invite you to submit a revised version of the manuscript that addresses the points raised during the review process.

We would appreciate receiving your revised manuscript by Nov 28 2019 11:59PM. To enhance the reproducibility of your results, we recommend that if applicable you deposit your laboratory protocols in protocols.io, where a protocol can be assigned its own identifier (DOI) such that it can be cited independently in the future. For instructions see: http://journals.plos.org/plosone/s/submission-guidelines#loc-laboratory-protocols

We look forward to receiving your revised manuscript.

Kind regards,

Tifei Yuan

Academic Editor

PLOS ONE

Journal Requirements:

2. We note that your paper includes detailed descriptions of individual patients/participants. As per the PLOS ONE policy (http://journals.plos.org/plosone/s/submission-guidelines#loc-human-subjects-research) on papers that include identifying, or potentially identifying, information, the individual(s) or parent(s)/guardian(s) must be informed of the terms of the PLOS open-access (CC-BY) license and provide specific permission for publication of these details under the terms of this license. Please download the Consent Form for Publication in a PLOS Journal (http://journals.plos.org/plosone/s/file?id=8ce6/plos-consent-form-english.pdf). The signed consent form should not be submitted with the manuscript, but should be securely filed in the individual's case notes. Please amend the methods section and ethics statement of the manuscript to explicitly state that the patient/participant has provided consent for publication: “The individual in this manuscript has given written informed consent (as outlined in PLOS consent form) to publish these case details”.

Reviewers' comments:

Reviewer's Responses to Questions

**Comments to the Author**

1. Is the manuscript technically sound, and do the data support the conclusions?

Reviewer #1: Yes

Reviewer #2: Partly

2. Has the statistical analysis been performed appropriately and rigorously? 

Reviewer #1: I Don't Know

Reviewer #2: Yes

3. Have the authors made all data underlying the findings in their manuscript fully available?

Reviewer #1: Yes

Reviewer #2: Yes

4. Is the manuscript presented in an intelligible fashion and written in standard English?

Reviewer #1: Yes

Reviewer #2: Yes

5. Review Comments to the Author

Reviewer #1: This manuscript examined a single-subject pilot study of repeated noninvasive stimulation on the ventromedial prefrontal cortex, suggesting a cumulative effect of repeated tDCS on the valence processing of emotional stimuli.

The abstract summarized the primary outcomes of the study and is logical. The introduction contains three paragraphs, and the first paragraph illustrated the vital role of vmPFC on emotion processing and the different activity pattern of pleasant and unpleasant stimuli. The second paragraph demonstrated the single tDCS effect on vmPFC. It revealed an increased neuronal activation of pleasant stimuli after excitatory tDCS and vice versa after inhibitory tDCS. It then pointed out the inconsistent outcomes of consecutive tDCS stimulation and the rationale of conducting the new tDCS paradigm. The third paragraph showed the hypothesizes of the research.

For the method, authors clearly described tDCS paradigm and MEG procedure. However, it seems a bit unclear about the passive viewing procedure. I am a bit confused whether the passive viewing task uses the same pictures in each tDCS sessions, or different pictures in each session? How the valence of pictures balanced? Besides, the study design without a sham tDCS group, how to prevent the placebo effect of the subject, especially under the non-blind condition?

The result is answered the three hypothesizes, and the three figures also well displayed the outcomes.

In the discussion part, the authors explained the relationship between three occurred cluster and tDCS stimulation, pointing out the possible changes of the neuroplasticity. Author also depicted potential limitations, especially the non-blind condition.

Reviewer #2: Comments to the Corresponding Author

The manuscript focuses on the cumulative emotional modulation of tDCS. The theme is worth studying and the manuscript makes good reading. However, there are several issues of concern that need to be addressed:

1) One big problem is the reliability and repeatability of the result. All findings were from a case, lacking of control condition that limits the interpretation and conclusions about the clinical significance of the results. The interpretation of the results is speculative, as mentioned in the limitations, these might be explained by inter-individual differences, and need to be tested in a representative sample.

2) The introduction seems to be too short and too vague. The authors need to provide a more comprehensive theoretical and empirical literature review of the associations between emotional modulation and repeated tDCS to highlight the importance of repeated tDCS in emotional processing research.

3) The interval between sessions of inhibitory stimulation and sessions of excitatory stimulation seems too long (one month), why?

4) How long did each stimulus present? How long is the interval between stimuli? More details are needed to describe the viewing task in Procedure.

5) Could you give further elaboration of results? It may be more readable and clearer to combine the analysis and result part.

6) In the discussion, author mentioned “While such implicit effects might potentially explain the non-cumulative effects across sessions it is unlikely to explain the cumulation of effects”, can you give a clearer explanation?

7) The authors admit the limitation of the study in that non-blinding design may cause additional influence such as attention. What ways do the authors consider can be used to examine or control this influence? Also, in the further study, single-blind or double-blind design might be more proper.

In sum, I had upon carefully reading this paper, though the research topic is of interest to many researchers in emotional modulation as well as in brain stimulation, but it needs much improvement.

6. PLOS authors have the option to publish the peer review history of their article (what does this mean?). If published, this will include your full peer review and any attached files.

Reviewer #1: No

Reviewer #2: No

---

## [Author Response · Author response to Decision Letter 0]

26 Dec 2019

Please see the attached document: Response to Reviewers.doc

---

## [Editor Report · Decision Letter 1]

6 Jan 2020

Repeated noninvasive stimulation of the ventromedial prefrontal cortex reveals cumulative amplification of pleasant compared to unpleasant scene processing: a single subject pilot study

PONE-D-19-23160R1

Dear Dr. Junghoefer,

We are pleased to inform you that your manuscript has been judged scientifically suitable for publication and will be formally accepted for publication once it complies with all outstanding technical requirements.

With kind regards,

Tifei Yuan

Academic Editor

PLOS ONE
---

## [Editor Report · Acceptance letter]

13 Jan 2020

PONE-D-19-23160R1 

Repeated noninvasive stimulation of the ventromedial prefrontal cortex reveals cumulative amplification of pleasant compared to unpleasant scene processing: a single subject pilot study 

Dear Dr. Junghofer:

I am pleased to inform you that your manuscript has been deemed suitable for publication in PLOS ONE. Congratulations! Your manuscript is now with our production department. 

With kind regards,

on behalf of

Dr. Tifei Yuan 

Academic Editor

PLOS ONE